# Prism Adaptation Treatment Predicts Improved Rehabilitation Responses in Stroke Patients with Spatial Neglect

**DOI:** 10.3390/healthcare10102009

**Published:** 2022-10-12

**Authors:** Tomas Vilimovsky, Peii Chen, Kristyna Hoidekrova, Ondrej Slavicek, Pavel Harsa

**Affiliations:** 1Department of Psychiatry, First Faculty of Medicine, Charles University, 121 08 Prague, Czech Republic; 2Kessler Foundation, West Orange, NJ 07052, USA; 3Department of Physical Medicine and Rehabilitation, Rutgers University, Newark, NJ 07101, USA; 4Department of Rehabilitation Medicine, First Faculty of Medicine, Charles University, 128 00 Prague, Czech Republic; 5Rehabilitation Center Kladruby, 257 62 Kladruby, Czech Republic; 6Department of Physiotherapy, Faculty of Physical Education and Sport, Charles University, 162 52 Prague, Czech Republic; 7Department of Informatics and Mathematics in Transport, Faculty of Transport Engineering, University of Pardubice, 532 10 Pardubice, Czech Republic

**Keywords:** spatial neglect, prism adaptation treatment, stroke, rehabilitation, motor recovery

## Abstract

Spatial neglect (SN) impedes functional recovery after stroke, leading to reduced rehabilitation gains and slowed recovery. The objective of the present study was to investigate whether integrating prism adaptation treatment (PAT) into a highly intensive rehabilitation program eliminates the negative impact of spatial neglect on functional and motor recovery. We examined clinical data of the 355 consecutive first-time stroke patients admitted to a sub-acute inpatient neurorehabilitation program that integrated PAT. The 7-item Motor Functional Independence Measure, Berg Balance Scale, and Motor Activity Log were used as main outcome measures. We found that 84 patients (23.7%) had SN, as defined by a positive score on the Catherine Bergego Scale via the Kessler Foundation Neglect Assessment Process (KF-NAP^®^). Although 71 patients (85%) received PAT, the presence of SN at baseline, regardless of PAT completion, was associated with lower functional independence, higher risk of falls, and a lower functional level of the affected upper limb both at admission and at discharge. The severity of SN was associated with inferior rehabilitation outcomes. Nonetheless, patients with SN who received PAT had similar rehabilitation gains compared to patients without SN. Thus, the present study suggests that integrating PAT in an intensive rehabilitation program will result in improved responses to regular therapies in patients with SN.

## 1. Introduction

Between 30% and 50% of stroke survivors suffer spatial neglect (SN) symptoms in the acute to subacute stage [1,2,3]. These patients fail to report, respond to, or orient to stimuli presented in the space contralateral to the injured cerebral hemisphere [4]. As a consequence, SN impedes functional recovery [5]. Patients with SN tend to have worse rehabilitation outcomes [3,6,7,8,9] and a slower rate of improvement during rehabilitation [6,10,11] compared to patients without SN.

Motor recovery is highlighted in stroke rehabilitation and is significantly affected by SN at different stages post-stroke. Chen et al. [10] assessed a cohort of stroke patients who were admitted to an inpatient rehabilitation program an average of six days post-stroke. Chen et al. [10] reported that in comparison with patients who did not show signs of SN, patients with SN had a lower level of functional independence, both at admission and at discharge, and stayed longer in the hospital. In addition, greater SN severity was associated with slower improvement of functional independence in the motor domain [10]. Similarly, Nijboer and colleagues [11] showed that greater SN severity is associated with less improvement in upper limb function, especially during the first 10 weeks post-stroke. Nijboer et al. [12] also reported negative impacts of SN on rehabilitation outcomes and motor recovery in patients who were admitted to a post-acute rehabilitation program an average of 56 days post-stroke. Katz et al. [6] observed that patients with SN had a lower level of functional independence at the time of admission (approximately one to two months post-stroke), at the time of discharge, and at the three-month follow-up evaluation compared to patients without SN, and patients with SN exhibited a slower recovery pattern. Cherney et al. [13] reported similar findings and observed a longer hospital stay among patients with SN. Hence, the problem observed in the literature is that SN decreases the effectiveness of inpatient rehabilitation care at both the acute and post-acute stages in addition to suppressing motor function recovery.

Integrating prism adaptation treatment (PAT) into standard rehabilitation care may reduce the adverse impact of SN on rehabilitation outcomes in U.S. inpatient rehabilitation facilities [14]. The promising finding [14] may not be generalized to other rehabilitation care systems. Langhamer et al. [15] compared nine specialized rehabilitation centers in seven different countries and found great disparities in length of stay, rehabilitation intensity, and therapeutic content, often with little reference to evidence-based practice. Even in developed countries in western Europe, among reputable rehabilitation facilities where evidence-based treatment is expected, variable therapeutic content and rehabilitation outcomes have been reported [16]. This wide variety in rehabilitation care practice also holds true in SN care. Significant differences have been reported in assessing [17] and treating SN [18] among facilities, professions, and rehabilitation care systems, which makes generalization of specific findings difficult. Thus, instead of generalizing published findings [14] to individual rehabilitation care systems, it is crucial to examine the impact of PAT on rehabilitation outcomes and motor recovery in systems that differ greatly from U.S. inpatient rehabilitation facilities.

The Czech Republic has quite a different inpatient rehabilitation care system for stroke survivors than the U.S. For example, in the Czech Republic, the length of stay is 10–12 weeks (much longer than in the U.S.) for all patients admitted to the intensive brain injury rehabilitation program, and patients are rarely discharged earlier. We recently implemented a set of evidence-based assessments and treatment of SN in such a brain injury rehabilitation center in the Czech Republic. The implementation was initiated as a prospective research study [19], and later evolved as a part of standard care in combination with high-intensity rehabilitation. To determine the benefit of adding PAT to the existing rehabilitation program, we conducted the present study using a retrospective chart review of real-world clinical data. Specifically, this observational study aimed to determine whether the integration of PAT into a high-intensity rehabilitation program predicted reduced adverse effects of SN to the extent that patients who presented with SN at the time admission were able to achieve a similar level of motor outcomes and functional recovery as patients without SN.

## 2. Materials and Methods

### 2.1. Patient Selection

This study was approved by the Institutional Review Board of the Rehabilitation Center (Kladruby, Czech Republic). We reviewed the medical records of patients admitted to the Intensive Brain Injury Rehabilitation Program at the Rehabilitation Center Kladruby (BIR Program) from June 2017 to July 2020. The BIR Program accepts patients who meet the following criteria: (1) are 18–75 years of age; (2) have an acquired brain injury; (3) have the potential to benefit from a minimum of 4 h of daily therapy in at least two of four different areas (psychology, occupational therapy, speech and language therapy, and physiotherapy), as determined by post-admission evaluations in each of these domains; (4) have an informal caregiver (e.g., family member) who will work with rehabilitation specialists during the inpatient stay; and (5) are expected to be discharged home.

### 2.2. Inclusion and Exclusion Criteria

From the available database of the BIR Program, we included patients with first-time stroke who had no prior brain injury documented in their admission medical record, in order to increase the homogeneity of the sample. We excluded patients with incomplete documentation or missing information. The final sample consisted of 355 patients.

### 2.3. Assessment for SN

All of the patients in the BIR Program received a comprehensive neuropsychological assessment upon admission. The assessment included several visuospatial tests that could reveal neglect signs and the Bells test, which specifically detects and measures SN [20]. The Bells test requires patients to use a pen or pencil to mark 35 bell-shaped targets among 280 non-targets printed on an A4 paper sheet (29.7 × 21 cm). If the neuropsychologist observed signs of SN in a patient during the assessment, then the patient was referred to an occupational therapist for additional functional assessment to confirm the diagnosis of SN.

Patients were assessed for SN by occupational therapists who followed the Kessler Foundation Neglect Assessment Process (KF-NAP^®^) for scoring on the Catherine Bergego Scale [21,22]. The KF-NAP is a highly sensitive measure [23,24], with good interrater reliability [23,25] and strong correlation with other tests [23,24], as well as other measures of functional status [3,23,24]. KF-NAP includes the following 10 categories: Limb awareness, personal belongings, dressing, grooming, gaze orientation, auditory attention, navigation, collisions, having a meal, and cleaning after a meal. Each category is scored from 0 (no neglect) to 3 (severe neglect) based on the therapist’s direct observation of patients in their hospital room during the morning hours before or after breakfast. The final score was calculated using the following formula: (sum score/number of scored categories) × 10 = final score [26]. The final score ranged from 0 to 30, and a positive score indicated the presence of SN. In the present study, patients were categorized as SN+ (KF-NAP > 0) or SN− (KF-NAP = 0). Within the SN+ group, patients were further categorized as mild neglect, i.e., mildSN+ group (KF-NAP = 1–10), or moderate-to-severe neglect, i.e., m-sSN+ group (KF-NAP ≥ 11).

### 2.4. Rehabilitative Therapies

The rehabilitation routines and therapy activities of the BIR Program are described elsewhere [19]. In addition to intensive standard care, the SN+ group received PAT. PAT was delivered using the treatment protocol and equipment of the Kessler Foundation Prism Adaptation Treatment [KF-PAT^®^] [27]. Each PAT session lasted 15–20 min and required the patient to perform 60 arm-reaching movements while wearing 20-diopter prism lenses that shifted the visual field to the ipsilesional side for 11.4 degrees of the visual angle.

### 2.5. Outcome Measures

The 7-item Motor Functional Independence Measure (7-item mFIM), Berg Balance Scale (BBS), and Motor Activity Log (MAL) were used routinely in the BIR Program for evaluation of functional and motor recovery rehabilitation success. We collected clinical data on these measures and calculated the change scores from admission to discharge as improvement indicators.

7-item Motor Functional Independence Measure [28]. The 7-item mFIM measures the level of independence. The FIM has excellent reliability [29,30], internal consistency [31], as well as adequate validity [30]. The BIR Program selected four items from the self-care subscale (eating, grooming, and dressing of both the upper and lower body) and three items from the transfers subscale (bed/chair/wheelchair, toilet, and tub/shower) as the standard measure for rehabilitation outcome. Each item is scored from 1 (maximal assistance) to 7 (complete independence). The total score ranges from 7 to 49.

Berg Balance Scale [32,33]. The BBS is a 14-item measure of static balance and fall risk. The BBS has excellent reliability, internal consistency, and validity in stroke population [34,35]. Each item is scored from 0 (inability to complete the item) to 4 (ability to independently complete the item). The total score ranges from 0 to 56.

Motor Activity Log [36].The MAL is a measure of a patient’s upper limb performance based on therapists’ observations. The MAL has shown excellent internal consistency and test retest reliability [37], as well as excellent criterion validity [38] in stroke population. A modified version was used in the BIR Program in which the therapist would examine only the number of successfully accomplished activities. The total score range is 0–30.

### 2.6. Analysis

Descriptive statistics and group comparisons were performed using the Mann–Whitney *U*-test for continuous variables or the chi-squared test for categorical variables. The impact of SN presence at admission on each of the rehabilitation outcome measures (improvement on 7-item mFIM, MAL and BBS) was examined using generalized linear models (GLMs) due to the heterogeneity of the sample (a common feature of clinical data). Besides the presence of SN at admission, the model also included rehabilitation outcome measures (7-item mFIM, MAL, and BBS) at admission; time post-stroke at admission; as well as sex, age, and years of education, in order to control for potential mediational effect of these factors. The same GLM structure was used to explore whether SN severity based on the classification of SN (moderate-to-severe vs. mild) at admission had an impact on rehabilitation gains. The significance level, or alpha, of all tests was set to 0.05, and *p*-values were based on two-sided tests. Analyses were performed with SAS 9.4.

## 3. Results

### 3.1. Impact of SN on Rehabilitation Outcomes

#### 3.1.1. Patient Characteristics

At the time of admission to the BIR Program, 84 of the 355 patients (23.7%) had SN (KF-NAP > 0). We excluded 10 patients from the analysis because they did not receive PAT, but rather an experimental sham treatment as they participated in a randomized sham-controlled trial [19], and another three because they did not receive any additional PAT treatment due to the unavailability of therapists. Thus, our final sample consisted of 71 patients with SN and 271 patients without SN. Both the SN+ and SN− groups were comparable in basic demographic characteristics such as age, years of formal education, time post-stroke, and the length of stay (Table 1). There was a statistical difference in the sex ratio; specifically, there were more females in the SN+ than the SN− group (*p* = 0.041). At the time of admission, the SN+ group had a lower score on the 7-item mFIM (*p* < 0.001), BBS (*p* < 0.001), and MAL of the affected upper limb (*p* < 0.001) compared to the SN− group (Table 1). Thus, the presence of SN was associated with inferior functional status at the time of admission.

#### 3.1.2. Rehabilitation Outcomes

The SN+ and SN− groups had similar lengths of stay (Table 1), which was expected, because the length of stay for stroke patients was pre-determined by insurers in the present healthcare system regardless of functional status at the time of admission. The presence of SN at the time of admission was associated with inferior scores on the BBS, 7-item mFIM and MAL at discharge (Table 1), although patients with SN received a varied number of PAT sessions (Table 2). Thus, at discharge, in comparison to the SN− group, the SN+ group required more support during ADL (indicated by the 7-item mFIM), had a higher risk of falling (BBS), and had lower function in the affected upper limb (MAL).

The results of each GLM on a given outcome measure are described below. Regarding the BBS gain, the model accounted for 56% of the variance, *F* (6, 303) = 64.15, *p* < 0.0001. For the model based on MAL gain, it explained 13.25% of the variance, *F* (6, 298) = 7.59, *p* < 0.0001. The model based on the 7-item mFIM gain explained 60.7% of the variance, *F* (6, 298) = 76.54, *p* < 0.0001. Table 3 summarizes the results of each model, indicating that the presence of SN did not independently predict improvements in any outcome variables after controlling for outcome at admission, time post-stroke at admission, years of education, age, and sex. Estimates of the average improvement value of each outcome measure based on each GLM are summarized in Table 4, showing that the rate of improvement between the two patient groups was statistically similar for the 7-item mFIM (*p* = 0.382), BBS (*p* = 0.600), and MAL (*p* = 0.259).

#### 3.1.3. Impact of SN Severity on Rehabilitation Outcomes

Fifty-four of the 71 patients (76.1%) with SN were classified into the mildSN+ group, and 16 patients (22.5%) were placed into the m-sSN+ group. One patient with SN was excluded from this analysis, as the data about the severity of SN were missing in the medical record. While receiving a greater number of PAT sessions (Table 2), the m-sSN+ group had worse rehabilitation outcomes compared to the mildSN+ group on all measures (Table 5).

The GLM on the BBS gain explained 61.31% of the variance, *F* (6, 62) = 16.37, *p* < 0.0001. The model on the MAL gain explained 18.16% of the variance, *F* (6, 62) = 2.29, *p* = 0.0461. The model on the 7-item mFIM explained 68.13% of the variance, *F* (6, 62) = 22.09, *p* < 0.0001. None of the outcome gains could be independently explained by SN severity after controlling for outcome at admission, time post-stroke at admission, years of education, age, and sex (Table 6). The results of the calculated estimates of the average improvement post-GLM (Table 6) indicated that both groups experienced similar improvement rates in the 7-item mFIM (*p* = 0.901), BBS (*p* = 0.167), and MAL (*p* = 0.120) (Table 7).

## 4. Discussion

The present observational study showed that motor function in patients with SN improved to a similar extent as patients without SN. Thus, addressing SN during an inpatient rehabilitation program has the potential to facilitate motor and functional recovery after stroke. This finding, however, is in contrast with previous findings in which SN slowed functional recovery [6,10,11]. Although patients with SN did not reach the same level of outcomes as patients without SN in the present study, the BIR Program with additional PAT [19] may have facilitated rehabilitation gains and led to partial removal of SN-related barriers to functional and motor recovery [5,11].

Another important finding was that people with moderate-to-severe SN responded to intensive rehabilitative therapies to a similar extent as people with mild SN. This finding is again inconsistent with previous reports that suggest an association between greater severity of SN and poorer rehabilitation improvements [10,13]. Thus, enhancing SN care in inpatient rehabilitation may be potentially beneficial. The BIR Program, from which clinical data were extracted for the present study, was intensive, comprising four to five therapy hours per treatment day for 10–12 weeks. The inpatient care was provided by a multidisciplinary therapy team that had access to the latest therapy devices and many evidence-based treatment protocols, including PAT (see [19] for a description). While we did not have information to systematically examine specific elements of the BIR Program that facilitated successful rehabilitation improvements in patients with SN, the findings of the present study are consistent with previous studies, which showed that implementing PAT into rehabilitation facilitated functional recovery [14], and a combination of different interventions may be more effective than a single intervention method for treating SN [39].

The rehabilitation care system, in which the present study was conducted, is a specific clinical setting that differs from other care systems. The standard care provided in the BIR Program involves not only physical and occupational therapy, as many other rehabilitation programs offer, but also psychological, neuropsychological, and speech and language therapy that is not considered optional, which is uncommon elsewhere [15]. The intensity of therapy in the BIR Program is also significantly greater than many other systems, because the evidence suggests that higher rehabilitation intensity leads to better recovery outcomes [40,41], which is particularly relevant to patients who are beyond two to three months post-stroke [42], the time frame for most patients admitted to the BIR program. Most studies have been conducted in settings where 2–3 h of therapy are provided daily [6,10,11,43], which is approximately one-half the intensity of the therapy received by patients in the BIR Program (4–5 h). In addition, patients usually stay in the BIR Program for 10–12 weeks. As shown in the present study, the median length of stay was 74 days, which is considerably longer than the 23 days reported in Chen et al.’s study [14], reflecting the difference in length of rehabilitation stay between the United States (17–23 days [10,15,34]) and European countries (45–75 days [16,44]). However, a longer hospital stay may not necessarily lead to better functional outcomes [45,46]; thus, it is unknown whether the length of hospital stay plays an important role. The cohort in the present study appeared to be 10–15 years younger than those in some previous studies [10,12,14]. Age plays a role in stroke rehabilitation outcomes, and the younger stroke population admitted to the BIR Program may have contributed to the discrepancy between the findings of the present study and those of previous studies. Future studies should evaluate the potential role of these factors.

SN impedes rehabilitation outcomes even after an intensive therapy regime. In the present study, the presence of SN was adversely associated with functional and motor recovery status at the time of admission and predicted inferior functional and motor recovery status at the time of discharge from the BIR Program, which was in agreement with previous studies [6,9,10,14]. Thus, although patients with SN had the same improvement as patients without SN, patients with SN did not achieve a similar level of functional or motor recovery as patients without SN. Knowing the negative consequences of SN in individuals’ functional independence [2,38,39] and life satisfaction [47], as well as the significant burden that SN represents for the healthcare system [48] and family members [2,49], the findings of the present study highlight how crucial it is to implement research-informed strategies that target SN during rehabilitation.

## 5. Study Limitations

One of the limitations of this study is the patient selection bias. Patients were first screened by a neuropsychologist using paper-based neuropsychological tests; however, those tests are not as sensitive as ecological assessment methods [24,50]. This approach may explain why the occurrence rate of SN in the present study was lower than that of some other studies [3,10]. There might be some patients classified as not having SN who could have shown signs of SN if assessed using the KF-NAP. Another limitation is that patients were not assessed for SN at the time of discharge. We had no information to comment on improvements in SN or the relationship with other outcome measures.

Another limitation of this study is that we did not have information on subtypes of spatial neglect and brain lesion sites. SN is a very heterogenous disorder, and some studies have shown that PAT is not equally effective for all patients [51,52,53,54]. Thus, including this information in the analysis could potentially provide some interesting insights.

The study findings are specific to a clinical setting where patients meet specific admission criteria and receive intensive inpatient rehabilitation care. Together, these features limit generalizability, but stakeholders are encouraged to consider implementing inpatient care with high intensity, rich resources, and accessibility to researchers.

## 6. Conclusions

SN has adverse effects on rehabilitation outcomes. Addressing SN during rehabilitation may facilitate rehabilitation gains. Despite the same improvement rates, patients with SN in the present study reached lower levels of functional and motor recovery outcomes than patients without SN. Therefore, we recommend further efforts in developing and implementing therapeutic treatments and interventions for patients with SN.

## Figures and Tables

**Table 1 healthcare-10-02009-t001:** Patient characteristics.

Variable	Patients with No Spatial Neglect (*n* = 271)	Patients with Spatial Neglect (*n* = 71)	*p*-Value of a Two-Group Comparison
Gender: female	94 (34.7%)	34 (47.9%)	0.041
Age (in years)	*n* = 271; 56 (48–64)	*n* = 71; 55 (48–62)	0.528
Education (in years)	*n* = 246; 12 (11–13)	*n* = 70; 12 (11–12)	0.515
Time post-stroke at admission (in days)	*n* = 271; 41 (27–61)	*n* = 71; 47 (28–66)	0.305
Length of stay (in days)	*n* = 271; 74 (54–79)	*n* = 71; 75 (63–80)	0.072
BBS at admission	*n* = 271; 36 (19–48)	*n* = 71; 16 (6–38)	<0.001
BBS at discharge	*n* = 263; 52 (46–56)	*n* = 71; 47 (33–53)	<0.001
7-item mFIM at admission	*n* = 270; 37.5 (31–42)	*n* = 71; 30 (24–36)	<0.001
7-item mFIM at discharge	*n* = 258; 42 (40–45)	*n* = 71; 41 (37–42)	<0.001
MAL at admission	*n* = 267; 12 (0–28)	*n* = 71; 0 (0–15)	<0.001
MAL at discharge	*n* = 258; 26 (4–29)	*n* = 71; 11 (0–27)	<0.001

Notes: Values are presented as the median (IQR) or count (%). Two groups were compared using the Mann–Whitney *U*-test for continuous variables or the chi-squared test for categorical variables.

**Table 2 healthcare-10-02009-t002:** Frequency of PAT sessions.

Number of PAT Sessions	Number of Patients (%)	Number of Patients in the mildSN+ Group (%)	Number of Patients in the m-sSN+ Group (%)
2	1 (1.4%)	0 (0%)	1 (6.3%)
3	1 (1.4%)	1 (1.9%)	0 (0%)
5	21 (29.6%)	21 (38.9%)	0 (0%)
9	1 (1.4%)	0 (0%)	1 (6.3%)
10	33 (46.5%) *	20 (37%)	12 (75%)
15	7 (9.9%)	7 (13%)	0 (0%)
20	7 (9.9%)	5 (9.2%)	2 (12.5%)
Total	71 (100%)	54 (100%)	16 (100%)

Notes: mildSN+ group refers to patients categorized as mild neglect (KF-NAP = 1–10), and m-sSN+ group refers to patients categorized as moderate-to-severe neglect (KF-NAP ≥ 11). * One patient with SN was not classified in any of the groups, as information about SN severity was missing from the medical records.

**Table 3 healthcare-10-02009-t003:** Parameter estimates of the GLM (patients with SN vs. patients without SN).

Outcome Variable	Effect	Coefficient	Standard Error	95% CI	*p*
BBS Gain	Spatial neglect (with SN)	0.63	1.20	−1.73; 2.99	0.600
	BBS at admission	−0.54	0.03	−0.60; −0.49	<0.001
	Time post-stroke at admission	−0.12	0.01	−0.15; −0.10	<0.001
	Education	−0.07	0.19	−0.45; 0.31	0.714
	Age	−0.13	0.04	−0.21; −0.04	0.004
	Sex (male)	1.58	1.01	−0.41; 3.57	0.119
MAL Gain	Spatial neglect (with SN)	−1.06	0.94	−2.90; 0.78	0.259
	MAL at admission	−0.17	0.03	−0.23; −0.11	<0.001
	Time post-stroke at admission	−0.04	0.01	−0.06; −0.02	<0.001
	Education	0.17	0.15	−0.13; 0.46	0.270
	Age	−0.05	0.04	−0.12; 0.02	0.142
	Sex (male)	−0.54	0.79	−2.10; 1.02	0.500
7-item mFIM Gain	Spatial neglect (with SN)	0.47	0.54	−0.59; 1.53	0.382
	7-item mFIM at admission	−0.56	0.03	−0.61; −0.50	<0.001
	Time post-stroke at admission	−0.04	0.01	−0.05; −0.03	<0.001
	Education	0.09	0.09	−0.08; 0.26	0.289
	Age	−0.04	0.02	−0.08; −0.00	0.033
	Sex (male)	0.32	0.45	−0.57; 1.21	0.477

Notes: Reference category of sex is female. Reference category of SN is “without SN”.

**Table 4 healthcare-10-02009-t004:** Estimated least-squares means after the general linear modeling for different patient groups based on the diagnosis of SN.

			95% Confidence Interval		
Outcome Variable	Group	LS Mean	Lower	Upper	*p*
BBS Gain	Patients with SN	15.79	13.73	17.84	
	Patients with no SN	15.16	14.03	16.28	
	Difference (with SN–without SN)	0.63	−1.73	2.99	0.600
MAL Gain	Patients with SN	4.35	2.75	5.95	
	Patients with no SN	5.41	4.52	6.30	
	Difference (with SN–without SN)	−1.06	−2.90	0.78	0.259
7-item mFIM Gain	Patients with SN	7.52	6.60	8.44	
	Patients with no SN	7.05	6.54	7.56	
	Difference (with SN–without SN)	0.47	−0.59	1.53	0.382

Notes: SN = spatial neglect.

**Table 5 healthcare-10-02009-t005:** Outcomes of patients with SN.

Variable	Mild SN (*n* = 54)	Moderate-to-Severe SN (*n* = 16)	*p*-Value of a Two-Group Comparison
Length of stay (in days)	76 (59–80)	74.5 (70–81.5)	0.499
BBS at discharge	49 (37–54)	32 (21.5–45.5)	0.004
7-item mFIM at discharge	41 (39–42)	38 (33–40.5)	0.008
MAL at discharge	13.5 (0–27)	0 (0–6.5)	0.005

Notes: Values are presented as the median (IQR) or count (%). Two groups were compared using the Mann–Whitney *U*-test for continuous variables or the chi-squared test for categorical variables.

**Table 6 healthcare-10-02009-t006:** Parameter estimates of the GLM (patients with mild SN vs. patients with moderate-to-severe SN).

Outcome Variable	Effect	Coefficient	Standard Error	95% CI	*p*
BBS Gain	BBS at admission	−0.60	0.07	−0.74; −0.46	<0.001
	Time post-stroke at admission	−0.15	0.03	−0.20; −0.10	<0.001
	SN severity (moderate-to-severe)	−3.74	2.68	−9.09; 1.61	0.167
	Education	−0.82	0.50	−1.81; 0.18	0.105
	Age	−0.44	0.10	−0.65; −0.23	<0.001
	Sex (male)	3.89	2.34	−0.78; 8.56	0.101
MAL Gain	MAL at admission	−0.07	0.08	−0.22; 0.08	0.362
	Time post-stroke at admission	−0.04	0.02	−0.08; −0.01	0.014
	SN severity (moderate-to-severe)	−2.82	1.79	−6.39; 0.75	0.120
	Education	−0.48	0.35	−1.17; 0.21	0.170
	Age	−0.12	0.07	−0.27; 0.02	0.093
	Sex (male)	1.00	1.57	−2.13; 4.13	0.525
7-item mFIM Gain	7-item mFIM at admission	−0.61	0.06	−0.73; −0.48	<0.001
	Time post-stroke at admission	−0.03	0.01	−0.05; −0.02	<0.001
	SN severity (moderate-to-severe)	0.14	1.10	−2.06; 2.33	0.901
	Education	−0.04	0.19	−0.41; 0.33	0.823
	Age	−0.15	0.04	−0.22; −0.07	<0.001
	Sex (male)	1.41	0.86	−0.31; 3.12	0.107

Notes: Reference category of sex is female. Reference category of SN severity is “mild”.

**Table 7 healthcare-10-02009-t007:** LS means for the category of neglect severity (patients with mild SN vs. patients with moderate-to-severe SN).

			95% Confidence Interval		
Outcome Variable	Group	LS Means	Lower	Upper	*p*
BBS Gain	Moderate-to-severe	17.01	12.40	21.62	
	Mild	20.75	18.31	23.20	
	Difference (moderate-to-severe–mild)	−3.74	−9.09	1.61	0.167
MAL Gain	Moderate-to-severe	2.92	−0.17	6.01	
	Mild	5.73	4.07	7.39	
	Difference (moderate-to-severe–mild)	−2.82	−6.39	0.75	0.120
7-item mFIM Gain	Moderate-to-severe	9.61	7.75	11.47	
	Mild	9.47	8.53	10.41	
	Difference (moderate-to-severe–mild)	0.14	−2.06	2.33	0.901

Notes: SN = spatial neglect.

## Data Availability

The data that support the findings of this study are available from the corresponding author upon reasonable request.

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
