# Peer review of "Prism Adaptation Treatment Predicts Improved Rehabilitation Responses in Stroke Patients with Spatial Neglect"

_healthcare, 2022, doi:10.3390/healthcare10102009_

Round 1
Reviewer 1 Report
In this paper, the authors evaluate the outcomes of including prism adaptation treatment (PAT) in a highly intensive rehabilitation program for treating functional and motor recovery in spatial neglect patients.
The study is clearly written and well-described and the results are interesting.
Nonetheless, I have some suggestions that could be relevant to the aims of the study. In particular, the authors overlooked the fact that neglect syndrome is a very heterogeneous pathology. In this regard, different studies have demonstrated that depending on the extent and location of lesions and the concomitance of primary sensory or motor deficits, the sensory, motor, and motivational components of the neglect syndrome attention can be differently affected in different patients.
see for example:
M. Aiello, S. Merola, F. Doricchi, Small numbers in the right brain: evidence from patients without and with spatial neglect, Cortex 49 (1) (2013) 348–351.
P. Bartolomeo, Attention Disorders After Right Brain Damage: Living in Halved Worlds, Springer Science & Business Media, 2013.
F. Doricchi, M.T. de Schotten, F. Tomaiuolo, P. Bartolomeo, White matter (dis)connections and gray matter (dys) functions in visual neglect: gaining insights into the brain networks of spatial awareness, Cortex 44 (8) (2008) 983–995.;
F. Lecce, F. Rotondaro, S. Bonni, A. Carlesimo, M.T. De Schotten, F. Tomaiuolo, F. Dorricchi, Cingulate neglect in humans: disruption of contralesional reward learning in right brain damage, Cortex 62 (2015) 73–88.
Based on these studies, one could consider that:
1) PAT rehabilitation programme could be highly effective in some particular forms of neglect rather than in other forms, and as a consequence...
2) I believe that the GLM analysis of rehabilitation outcomes could be greatly improved by considering not only neglect severity but also neglect typology (e.g. personal, motor, extra-personal, representational etc...). In a similar way, entering lesion sites and lesion extensions within covariates could provide interesting insights.
Reviewer 2 Report
Title: Prism Adaptation Treatment Predicts Improved Rehabilitation Responses in Stroke Patients with Spatial Neglect
In this study, the authors investigate whether integrating prism adaptation treatment (PAT) into a highly intensive rehabilitation program eliminates the negative impact of spatial neglect on functional and motor recovery. This study has chosen a good topic of research in Patients with Stroke. With all humility, these recommendations are collected with the intention that they can be of help to improve this work.
Introduction
The introduction is so long, it is suggested to make a more in-depth review since there are only six citations in Spatial Neglect. It is recommended:
- Rather than listing the research results of the six papers related to Spatial Neglect, it would be better to summarize Spatial Neglect by focusing on problems and treatment methods.
2.1. Patient Selection: Separately describe selection conditions and exclusion conditions.
2.2. Assessment for SN: Please add the reliability and validity of the measurement tool
2.4. Outcome Measures: Please add the reliability and validity of the measurement tool
Thanks.
Round 2
Reviewer 1 Report
I understand the problem raised by the authors in missing data for the additional analysis I requested in the previous revision.
Adding such a problem to the limitations of the present study could be considered a good compromise.